

# Comparison of two benthic assemblage sampling gears for use on intertidal oyster reefs in Louisiana

Finella M. Campanino[1], Stephanie K. Archer[2], Jillian C. Tupitza[3], Cassandra N. Glaspie[3] and Megan K. La Peyre[4]

[1] School of Renewable Natural Resources, Louisiana State University Agricultural Center, Baton Rouge, Louisiana, United States
[2] Louisiana Universities Marine Consortium, Chauvin, Louisiana, United States
[3] Department of Oceanography and Coastal Sciences, Louisiana State University and Agricultural and Mechanical College, Baton Rouge, Louisiana, United States
[4] U.S. Geological Survey, Louisiana Cooperative Fish and Wildlife Research Unit, School of Renewable Natural Resources, Louisiana State University Agricultural Center, Baton Rouge, Louisiana, United States

Corresponding author
Megan K. La Peyre,
mlapeyre@agcenter.lsu.edu

## ABSTRACT

**Background:** Estuarine biodiversity plays a vital role in supporting ecosystem functions yet remains threatened by climate change and anthropogenic activity. Tracking and identifying estuarine biodiversity trends helps management ensure long-term provisions of human and environmental benefits by contributing to the estimation of habitat loss and the monitoring of restoration and conservation progress. However, results obtained using different sampling gears and different biodiversity metrics may lead researchers to reach different conclusions, which can lead to uncertainty in the actual state of the ecosystem-level biodiversity. Sampling benthic biodiversity in complex estuarine habitats, such as oyster reefs, is particularly challenging because no one gear type captures entire target assemblages, and differences in gear efficiency on these complex habitats make comparisons across gear types challenging.

**Methods:** We investigated how estimates of oyster reef-associated benthic taxa abundance, richness, Pielou's evenness, and Shannon-Wiener diversity differed across three *Crassostrea virginica* reefs in Louisiana between suction sampler and substrate tray sampling gears ($n = 6$), and how gear influenced comparisons across reefs (3 reefs × 6 replicates × 2 gears).

**Results:** Abundance and richness were higher, and Pielou's evenness was lower, in trays compared to suction samples at all reefs. Shannon-Wiener diversity was similar in suction samples and trays at two out of three reefs. Amphipod taxa were numerically dominant in trays, skewing the distribution of abundances and driving the reef assemblage differences between gears. Abundance and Shannon-Wiener diversity were similar across reefs within each gear. However, there were significant differences in richness across reefs in tray samples only, while evenness differed across reefs only in suction samples. Our results highlight that gear choices, along with biodiversity metrics tracked, can result in different conclusions in biodiversity trends, ultimately affecting conservation decisions and management.

# INTRODUCTION

Loss of estuarine biodiversity due to climate change and anthropogenic activities may negatively impact estuarine habitats' provisioning of ecosystem functions and services (*Lotze et al., 2006*). To address this loss, many local and global initiatives focus on preserving, restoring, and enhancing ecosystems to maintain biodiversity in estuaries (*United Nations Environment Programme (UNEP), 2021*). The identification and development of robust monitoring tools and metrics enable tracking and understanding of the impacts of these management efforts on biodiversity (*Flannery & Przeslawski, 2015*).

Monitoring biodiversity in estuaries is challenging due to the diverse and often structurally complex habitats (*e.g.*, shellfish reefs) impacting sampling gear efficiency (*Baker & Minello, 2011*; *Flannery & Przeslawski, 2015*; *Mihoub et al., 2017*; *La Peyre et al., 2021*). The use of different gears to sample estuarine benthic assemblages (hereafter assemblages), such as ponar grabs, trays, or suction samplers can result in different values for taxa abundances, assemblage structure, diversity, and richness (*Stoner et al., 1983*; *Slack, Ferreira & Averett, 1986*; *Keklikoglou et al., 2019*). One of the more complex habitats within estuaries are reefs built by the Eastern oyster *Crassostrea virginica*. These reefs support significant biodiversity but estimates of assemblage abundance and richness vary between studies that have sampled reefs of different complexity and/or used different gears (*Wells, 1961*; *Coen & Grizzle, 2007*; *La Peyre et al., 2019*). For example, a meta-analysis of studies from coastal Texas through Florida documented 115 fish and 41 decapod crustacean species with densities as high as 2,800 m$^{-2}$ and as many as 52 species per reef, but the results differed significantly based on the gear types used (*La Peyre et al., 2019*). These studies highlight that gear comparison is complicated by habitat characteristics which can affect a gear's ability to capture a representative proportion of the target assemblages (efficiency) and specific taxa and size classes (selectivity) across habitats (*Wells, 1961*; *Stoner et al., 1983*; *Slack, Ferreira & Averett, 1986*; *Coen & Grizzle, 2007*; *Keklikoglou et al., 2019*; *La Peyre et al., 2019*). Consequently, researchers and managers still search for gear that maximize the taxa richness captured, minimize sampling time and effort, and promote more consistent biodiversity assessment across habitats, locations, and seasons (*Flannery & Przeslawski, 2015*).

To date, there are no studies in oyster reefs comparing the selectivity and efficiency of trays and suction sampler gears designed to sample benthic assemblages. Trays are often used for sampling biodiversity on oyster reefs (*La Peyre et al., 2019*). They create minimal reef disturbance, minimize loss of escaping organisms, and allow the collection of organisms that reside in reef interstitial spaces (*Beck & La Peyre, 2015*). However, trays are time-consuming to deploy, have a high risk of gear loss due to long deployment times, and potentially bias results due to the added structure of the tray (*Beck & La Peyre, 2015*). In contrast, suction samplers require less time and have a lower chance of gear loss because samples are collected in one field event with no deployment time required. Suction

samplers are often limited to shallow water habitats and capturing size classes as large as the suction diameter. Additionally, suction samplers create a noise disturbance from the motor, which can impact biodiversity sampling by scaring away or altering behavior of noise-sensitive species (*Flannery & Przeslawski, 2015*). It is also unclear how efficiently suction samplers capture organisms that reside in the interstitial space within reef substrate. Until recently, the suction sampler was used primarily on soft bottom (*True, Reys & Delauze, 1968*) but is gaining popularity for sampling oyster reefs (*Pinnell et al., 2021*; *Pollack, Palmer & Williams, 2021*).

In this study, we compared two sampling gears, trays and suction samplers, in collecting benthic biodiversity on natural intertidal *C. virginica* oyster reefs. By comparing four common metrics of biodiversity calculated from each sampling gear across three reefs, we hypothesized that tray samples would consistently yield higher taxa abundance, richness, Pielou's evenness, and Shannon-Wiener diversity than suction samplers. This study provides insight into potential biases or limitations of each gear, leading to more accurate and comparable data for future studies. Comparing benthic biodiversity sampling gears informs management and conservation strategies for oyster reefs, which hold significant ecological and economic value.

## MATERIALS AND METHODS

### Study site

The study area encompassed approximately 1 km$^2$ of coastal habitats near Cocodrie, Louisiana which included *Spartina alterniflora* dominated marsh, microtidal channels, bayous, and ponds interspersed with oyster reefs (Fig. 1A). Over the past decade (2012–2022) water depth ranged from 1.0–2.5 m ($1.7 \pm 0.0009$, $n = 52,497$), water temperature ranged from 1.5–35.2 °C ($23.0 \pm 0.01$, $n = 309,313$), and salinity ranged from 0.6–25.1 PSU ($9.5 \pm 0.01$, $n = 241,430$; http://weatherstations.lumcon.edu/index.html).

### Sampling design

We selected three reef sites (30 m × 20 m; hereafter reef) located at least 100 m apart and centered over natural intertidal oyster reefs (coordinates: Reef 1 29.25284, −90.67362, Reef 2 29.25616, −90.66748, Reef 3 29.2566, −90.66621; Fig. 1B). At each reef, trays ($n = 6$) were deployed June 29, 2022, and collected July 11–12 and suction samples ($n = 6$) were collected July 14 to minimize reef disturbance from the collection of the trays. Temperature (°C), salinity (PSU), dissolved oxygen (mg L$^{-1}$), and water depth (m) were downloaded from the *Louisiana Universities Marine Consortium (2023)* environmental monitoring station referenced above, located within 1,000 m of all reefs, from July 11–14. All reefs were completely submerged where any samples were taken to minimize inconsistencies due to microtidal fluctuations and the mean depth of the center of the reefs was $0.72 \pm 0.17$ m.

### Field sampling

To characterize the habitat structure of each reef, six 0.25 × 0.25 m quadrats were haphazardly placed within the submerged reef and reef substrate was collected manually to

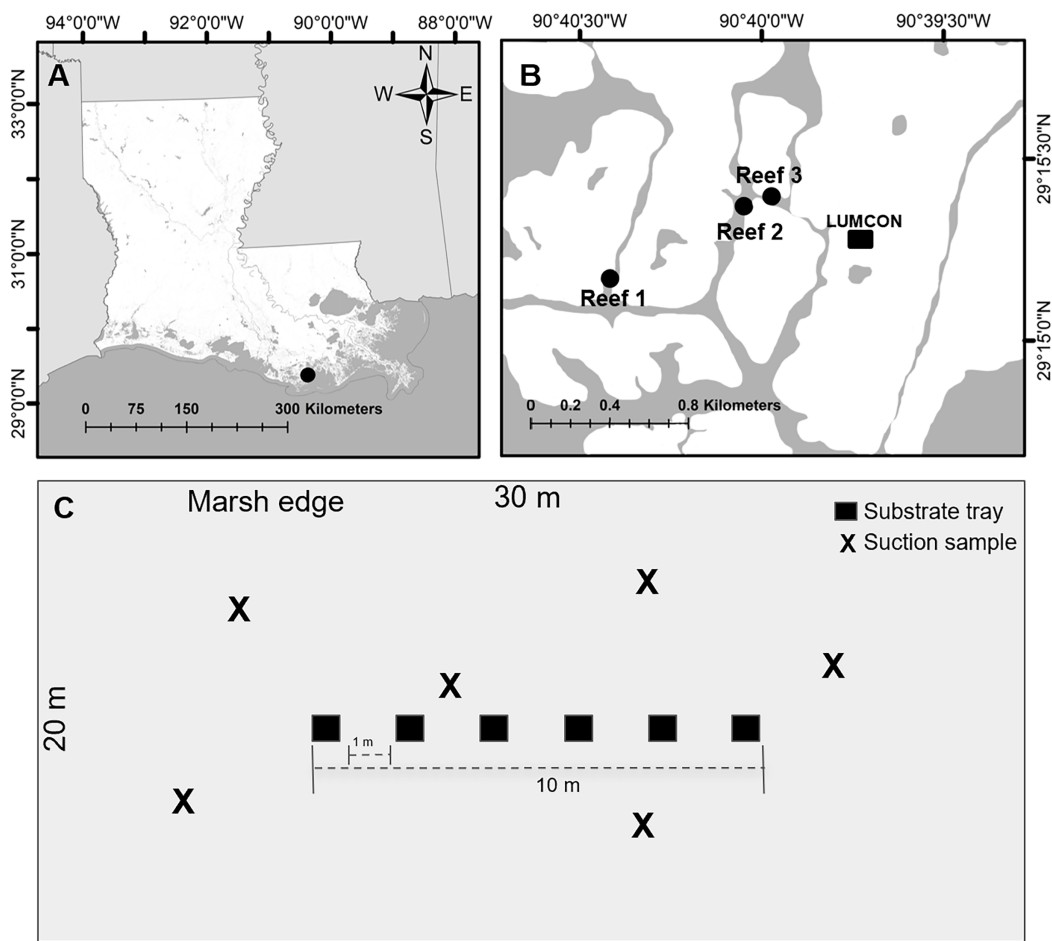

**Figure 1 Experimental design set up.** (A) Image of study area (black circle) in Louisiana. (B) Image of reef locations (black circles) within the study area. (C) Example of one reef where trays and suction samplers were located. Trays were placed centered on each reef and suction samples were taken haphazardly on each reef. Images created using ArcGIS Mapping Software with background from the *Multi-Resolution Land Characteristics Consortium (2016)*.

10 cm depth. For each quadrat, reef substrate volume (L; live + dead oyster material), reef cluster volume (L: all attached shells and oysters forming larger groups of shell material) as measures of complexity per (*Beck & La Peyre, 2015*), and abundance of the live oysters were recorded. Both volume measurements were estimated *via* water displacement.

Trays (0.48 m × 0.48 m × 0.10 m; 20 L) lined with 3-mm chicken wire and 1-mm mesh bags were deployed on reefs ~2 weeks prior to collection to allow the assemblages to develop (*Beck & La Peyre, 2015*). The trays were filled with ~3 L of oyster shells and topped with ~1.5 L of reef substrate taken from a single quadrat saved from the reef characterization (*Beck & La Peyre, 2015*). The trays were placed ~1 m apart in a row parallel to the marsh edge and centered on each reef (Fig. 1C). A lead line connected to the trays was secured to the marsh's edge using a PVC pole, ensuring relocation and recovery. At tray retrieval, the mesh bags lining the trays were cinched closed to reduce organism escape.

Suction samples were collected using a haphazardly placed 0.48 m × 48 m × 0.30 m throw trap that enclosed an identical benthic area as the tray (Fig. 1C). The throw trap was tossed onto the reef, and then pushed into the mud to ensure a proper seal before clearing with a suction sampler device (La Peyre et al., 2019). Using a gas-powered, venturi suction sampler device with a 10.16 cm suction diameter similar to Glaspie et al. (2018), we suctioned the reef bottom for 15 s based on trials examining the amount of time to suction the 0.48 m × 0.48 m bottom of the throw trap; based on preliminary trials using the suction sampler within a throw trap, it is estimated that the reef is suctioned to an average 10 cm depth, matching the depth of the trays sampled. The suctioned organisms and loose sediment were discharged into 1-mm mesh bags. All tray and suction samples were rinsed through a 1-mm sieve and placed in sample bags on ice until they were stored at −20 °C in the lab for later processing.

## Laboratory analyses

Each sample was thawed, and all benthic organisms were identified to the lowest practical taxonomic unit. All taxa were enumerated, and wet weights were recorded (g). For amphipods and isopods only, a maximum of 50 individuals per sample were identified and the remaining amphipods/isopods were grouped, and total wet weight biomass was recorded. All taxa were dried at 60 °C until a constant weight was achieved, and dry weight was recorded (0.0001 g; Beck & La Peyre, 2015). The grouped amphipod/isopod biomass was used in conjunction with the weights of the 50 identified amphipods and isopods to estimate the total number of amphipod and isopod taxa per sample (Eq. S1).

When a taxon not previously identified by this research team was found, their wet weight was recorded, and a voucher specimen was taken. The voucher's dry weight was estimated based on the mean dry weight of the other individuals from that taxon. This weight was then added back into the taxon-specific dry weight for that sample. The wet weight was used as a biomass estimate for dry weight once because only one representative of the voucher taxon was found. All voucher specimens were catalogued and added to LUMCON Natural History Collection.

## Statistical analysis

All taxa abundance and biomass per sample were divided by 0.2304, the surface area (m$^2$) of each tray/suction sample, to standardized estimates to 1 m$^2$ for comparison among similar studies. We chose to analyze abundance instead of biomass because our estimates of abundance and biomass were correlated (Spearman's correlation: r$_s$ = 0.79, $p < 0.001$; Table S1) and abundance m$^{-2}$ was reported more frequently across similar studies. Shannon-Wiener diversity was calculated based on abundance *via* the 'vegan' package (Eq. S2; Oksanen et al., 2022) and Pielou's evenness was calculated by dividing the Shannon-Wiener diversity by the log of richness. We tested for differences in taxa abundance, richness, Pielou's evenness, and Shannon-Wiener diversity within a reef between gears, and among reefs within a gear using Kruskal-Wallis tests because the data did not meet normality assumptions for parametric tests *via* the Shapiro-Wilk test. When comparing between gears we used separate tests for each reef because there were
reef-specific differences and a Bonferroni adjustment ($\alpha = 0.01$) to control for multiple comparisons. To compare among reefs within a gear, we used an $\alpha$ of 0.05 for the Kruskal-Wallis test and a Dunn test with a Bonferroni adjustment ($\alpha = 0.01$) for *post-hoc* comparisons. Statistical analyses were conducted in R version 4.1.0 (*R Core Team, 2022*). This study was performed under the auspices of Louisiana State University Agricultural Center protocol # A2021-08. Approval for field work was provided by Louisiana Universities Marine Consortium (LUMCON), verbally, by Brian Roberts.

## RESULTS

Throughout collection days water depth ($1.85 \pm 0.01$ m, $n = 403$), temperature ($29.15 \pm 0.04\,^{\circ}$C, $n = 384$), salinity ($5.96 \pm 0.06$ PSU, $n = 384$), and dissolved oxygen ($5.42 \pm 0.05$ mg L$^{-1}$, $n = 384$) were typical for this region during the summer (Table S2).

Across the reefs, reef substrate (live + dead oyster material) volume ranged from 4–48 L m$^{-2}$, live oyster densities ranged from 0–144 ind m$^{-2}$, and cluster volume ranged from 3.2–32 L m$^{-2}$ (Table S3). Reef 1 had the lowest live oyster density ($0 \pm 0$ ind m$^{-2}$) and substrate volume ($18.29 \pm 3.36$ L m$^{-2}$) while reef 2 had the highest live oyster density ($73.14 \pm 21.20$ ind m$^{-2}$) and substrate volume ($28.57 \pm 5.27$ L m$^{-2}$).

Trays consistently contained significantly higher abundances compared to suction samples (Kruskal Wallis tests: Reef 1 $\chi^2 = 8.34$, df = 1, $p = 0.004$; Reef 2 $\chi^2 = 8.31$, df = 1, $p = 0.004$; Reef 3 $\chi^2 = 8.34$, df = 1, $p = 0.004$) and no significant differences in abundance were detected among reefs for either gear (Fig. 2A). Similarly, trays consistently contained significantly higher richness than suction samples (Kruskal Wallis tests: Reef 1 $\chi^2 = 8.49$, df = 1, $p = 0.004$; Reef 2 $\chi^2 = 8.49$, df = 1, $p = 0.004$; Reef 3 $\chi^2 = 6.70$, df = 1, $p = 0.0096$; Fig. 2B). In trays, there was significantly lower richness at reef 1 compared to reef 2, and no significant differences between these reefs and reef 3 (*Post Hoc* Dunn tests: Reefs 1–2 Z = $-2.75$, $p$ adj. = 0.02). No significant differences in richness among reefs were detected from suction samples. Trays consistently contained significantly lower Pielou's evenness compared to suction samples across reefs (Kruskal Wallis tests: Reef 1 $\chi^2 = 7.52$, df = 1, $p = 0.006$; Reef 2 $\chi^2 = 8.37$, df = 1, $p = 0.004$; Site 3: $\chi^2 = 8.34$, df = 1, $p = 0.004$; Fig. 2C). In suction samples, there was significantly higher evenness at reef 1 compared to reef 2, and no significant differences between these reefs and reef 3 (*Post Hoc* Dunn tests: Reefs 1–2 Z = 2.58, $p$ adj. = 0.02). No significant differences in evenness among reefs were detected from tray samples. For tray and suction samples, Shannon-Wiener diversity did not differ significantly between gears at reefs 1 and 2 but was significantly higher in suction samples than trays at reef 3 (Kruskal Wallis test: $\chi^2 = 8.43$, df = 1, $p = 0.004$; Fig. 2D). However, no significant differences in Shannon-Wiener diversity were detected among reefs for either gear.

## DISCUSSION

The collection of comparable data across projects and locations enables the development of robust databases to inform biodiversity protection (*Flannery & Przeslawski, 2015*). Here, we compared suction samples and trays to assess benthic biodiversity metrics on and between oyster reefs. We found significant differences in biodiversity metrics between

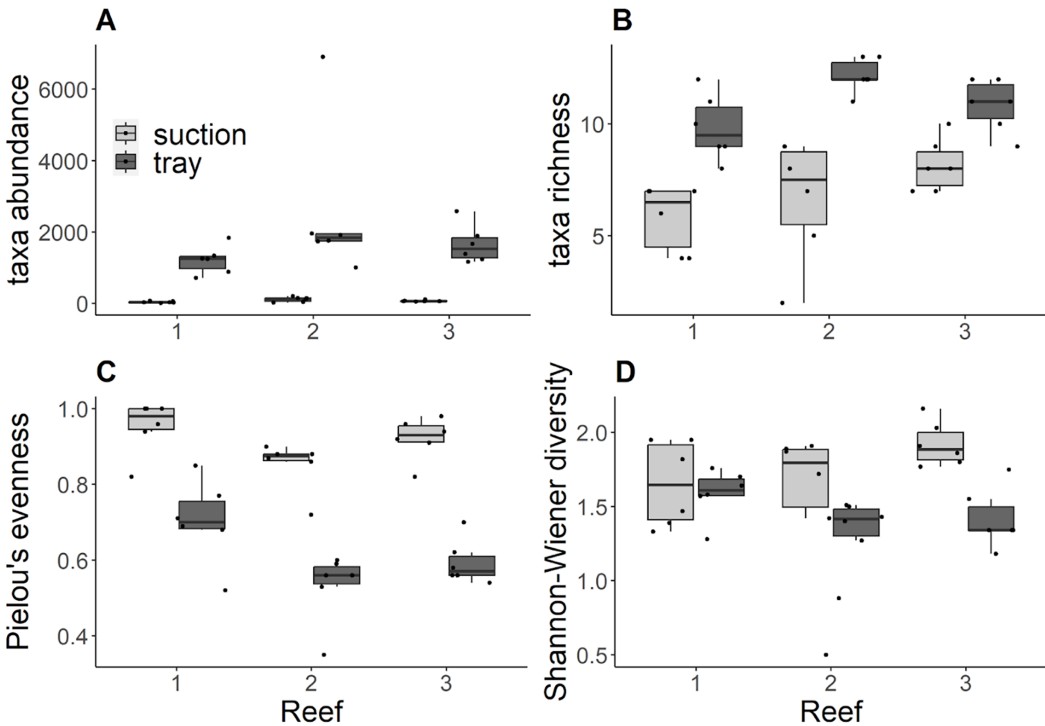

**Figure 2 (A) Taxa abundance ind. m⁻², (B) taxa richness, (C) Pielou's evenness, (D) Shannon-Wiener diversity by reef and grouped by sampling gear.** Black dots indicate each sample replicate, horizontal lines through the boxes indicate the median, lower and upper extents of the boxes correspond to the first and third quartiles (Q1 and Q3). The upper whisker represents Q3 × 1.5 × interquartile range (IQR) and the lower whisker represents Q1 × 1.5 × IQR.               

gears and across reefs that varied by metric. Gear selectivity and efficiency, which can differ by habitat characteristics, likely contribute to differences in gear outcomes, further indicating that tray and suction sampling are not comparable biodiversity sampling gears on oyster reefs. Using two gear types to compare selected biodiversity metrics across a complex estuarine habitat, we highlight the importance of understanding and acknowledging how sample gear efficiency, and selectivity may impact biodiversity estimates and comparisons. These findings may be beneficial to managers developing monitoring and adaptive management plans and help inform conservation decisions by acknowledging uncertainty or variation in biodiversity estimates resulting from sampling protocols and metric calculations.

Biodiversity metrics often differ based on the targeted assemblages collected. Here, the average organism abundance of 1,806 ind m⁻² (SE ± 318 m⁻² $n = 18$) and 23 total taxa collected in trays was higher than majority of average fish and/or decapod crustacean abundance (ranging from 57–1,579 ind. m⁻²) and total taxa (ranging from 8–22) per study that used trays as sampling gear in an oyster reef assemblage meta-synthesis across the U.S. Gulf Coast (USGC; *La Peyre et al., 2019*). Only two studies in the meta-synthesis, both from Texas reefs, reported higher numbers of abundance (2,856 ind m⁻²; *Rezek et al., 2017*) or total taxa (25; *Blomberg et al., 2018*). The meta-synthesis studies only included fish and/or decapod crustaceans, while here we also include smaller taxa such as

amphipods, isopods, and polychaetes. When removing the smaller taxa from this dataset, average abundance (698 ± 51 m$^{-2}$ $n = 18$) and total taxa (15) were more comparable to studies in *La Peyre et al. (2019)*. Additionally, tray modifications, such as lining trays with mesh drawstring bags, can impact gear efficiency. We lined trays with a smaller mesh size (1 mm) compared to past studies which likely contributes to the higher abundance observed (*Beck & La Peyre, 2015*). The inclusion of amphipods, which were numerically dominant in our trays along with our gear modification, likely drove the higher abundances and richness captured but does not explain lower suction sampling results in this study as other studies have largely focused on smaller fauna when suction sampling.

The use of suction sampling in reef environments in other studies is sparse but the assemblage metrics are similar to trays in the USGC likely because of the epifauna and infauna target assemblages (*La Peyre et al., 2019*; *Pinnell et al., 2021*; *Pollack, Palmer & Williams, 2021*). In contrast, our suction sample abundances ranged from 17–204 ind m$^{-2}$ and richness from 2–10 which are lower than the few similar studies' ranges of abundance (241– 8,800 ind m$^{-2}$) and richness (6–12) per suction sample (*Pinnell et al., 2021*; *Pollack, Palmer & Williams, 2021*). Differences in gear efficiency can create discrepancies across gears and is further impacted by gear modifications, such as modifying the amount of time used to suction a given area (*Brown, Schram & Brussock, 1987*). We suctioned 0.23 m$^{-2}$ area for 15 s, which is lower than 0.0625 m$^{-2}$ area for 30 s in *Pinnell et al. (2021)*, likely contributing to the lower abundance and richness in this study. Although *Pinnell et al. (2021)* collected organisms in a different geographic region (Pacific coast) compared to this study (USGC), the cutoff size (>500 μm *vs* >1 mm in this study) of organisms and gear modifications used likely drove the comparatively higher assemblage metrics. Additionally, *Pollack, Palmer & Williams (2021)* combined gears on Texas reefs, using suction samplers to suction reef habitat placed within trays, whereas we used suction samplers and trays separately. Differences in reported abundance and richness between studies may partially stem from variations in gear design and methods (gear combinations, mesh size), resulting in differences in gear selectivity.

Although there were many taxa caught by both sampling gears across reefs, suction samplers and trays captured some taxa that the other did not (tray = 8, suction = 6). Generally, suction samples captured more sessile invertebrates (*e.g.*, *Ameritella mitchelli*) associated with mud-bottom habitat than trays, while trays captured more mobile reef-associated taxa that suction samples did not (*e.g.*, *Gobiesox strumosus*, Table 1) across reefs. The average number of taxa captured only in trays (mean $_{richness}$ = 3.77 ± 0.68 m$^{-2}$, $n = 144$ $_{individuals\ per\ sample}$) were more abundant than average taxa captured only in suction samples (mean$_{richness}$ = 0.56 ± 0.17 m$^{-2}$, $n = 108_{individuals\ per\ sample}$). These gear-specific taxa contributed to the higher tray abundance and richness compared to suction samples and is logical based on how each gear operates. Our two-week tray deployment time was not long enough for sessile invertebrates to settle and grow large enough to identify. Additionally, mobile taxa can escape the enclosed suction sampling area on uneven oyster reef substrate, likely explaining the fewer mobile taxa caught. In suction sampling studies within seagrass, a drop net is used to prevent mobile taxa from escaping and is not inherently a limitation of suction sampling (*Ralph et al., 2013*). However, it appears the use of drop nets with

**Table 1 Scientific and common names of taxa that were only captured by either trays or suction samples.**

| Scientific name | Common name | Average abundance m$^{-2}$ |
|---|---|---|
| **Trays** | | |
| Diptera Chironomidae | Midge larvae | 2.41 |
| *Pachygrapsus gracilis* | Dark shore crab | 0.24 |
| *Ctenogobius boleosoma* | Darter Goby | 3.86 |
| *Gobiesox strumosus* | Skillet fish | 20.01 |
| *Hypsoblennius hentz* | Feather blenny | 1.69 |
| Isopoda Ancinidae | Isopod family | 0.72 |
| Nemertea | Ribbon worm | 0.48 |
| Alpheidae spp | Snapping shrimp | 0.72 |
| **Suction samples** | | |
| *Ampelisca abdita* | Amphipod spp | 0.96 |
| *Mytilopsis leucophaeata* | Conrad false mussel | 0.96 |
| *Ameritella mitchelli* | Mollusc spp | 0.24 |
| *Arcuatula papyria* | Atlantic paper mussel | 0.48 |
| Pleuronectiformes | Young of the year flatfish | 0.48 |
| *Gobionellus oceanicus* | Highfin goby | 0.24 |

suction sampling has not translated to reefs yet. Reporting gear modifications and standardized (m$^{-2}$) biodiversity metrics allows for the comparison of assemblages across reefs and gears for biodiversity monitoring.

Our findings indicate tray and suction sampling are not comparable within a single oyster reef nor across oyster reefs. One explanation for the differences across oyster reefs within a single gear may be due to differences in reef characteristics (*e.g.*, structural complexity), which can drive differences in the assemblage patterns among reefs (*Pinnell et al., 2021*). However, tray and suction samples detected contrasting differences among reefs for richness and evenness and different relative rankings across space (*e.g.*, tray average richness: reef 2 > reef 3 > reef 1, suction average richness: reef 3 > reef 2 > reef 1; Table S3). Additionally, when considering unique taxa (*i.e.*, a taxon represented by a single individual within a sample), on average 51% of the taxa collected in a suction sample were unique, whereas only 17% of taxa within a tray were unique. More unique taxa collected in suction samples was likely due to a lack of abundance of common taxa also collected in trays in higher abundance. All trays had at least one numerically dominant taxa whose abundance was more than two times the sample-specific mean, but this only occurred in 44% of suction samples. The lower proportion of unique taxa and higher proportion of numerically dominant taxa in trays here explain the differences in Pielou's evenness and Shannon-Wiener diversity observed between the gears. Contrasting gear results of richness and abundance with Pielou's evenness led to the similarity in Shannon-Wiener diversity between gears. The samples collected *via* trays consistently contained more individuals, more dominant taxa, and had significantly different patterns among reefs compared to

suction samples, highlighting the incomparable biodiversity metric results generated from these gears on oyster habitat.

## CONCLUSIONS

Sampling gear choice depends on factors such as time frame, equipment, funding, field standards, and target assemblages. Our study demonstrated that oyster reef-associated biodiversity metrics varied significantly between gears and across reefs. Trays captured more individuals and taxa overall, including dominant taxa, while suction samplers collected more unique taxa. While *Pollack, Palmer & Williams (2021)* captured taxa abundance and richness using suction samples that aligned with previous data captured using trays, we did not. The discrepancy highlights the potential influence of other factors, such as habitat complexity, area sampled, and suction duration, on estimates of biodiversity generated from suction and tray samples. Comparing and standardizing field methods, rather than continually developing new sampling approaches, may be more beneficial for reconciling discrepancies to future research and monitoring. Moreover, further research on the impacts of gear modifications, environmental factors, and habitat characteristics may enhance sampling gear decision making. Understanding biodiversity metrics and associated trade-offs for selecting a gear remains critical because the gear's effectiveness is dependent on the project goals and target assemblages (*Yi et al., 2012*). Our study contributes to informed decision-making by managers, researchers, and practitioners about gear selection and data interpretation for targeted studies and monitoring programs.

## ACKNOWLEDGEMENTS

We would like to thank J. Bowman, A. Host, H. Crawford, G. Hancock, A. Rioux, D. McMahon, S. Hall, V. Hoff, A. Mehrotra, S. Liner, N. Coxe, M. Bates, D. Lambert, T. Davenport, S. Yongue, K. Schlachter, Z. Guo, E. Vellemarette, B. Farmer, A. Lipford, L. Moran, and A. Hill for their assistance in the field and lab. We thank A. Nyman for comments on an early draft of this manuscript. The present work was part of Finella Campanino's Master's thesis. Any use of trade, firm, or product names is for descriptive purposes only and does not imply endorsement by the U.S. Government.

### Funding

This work was funded by the Louisiana Department of Wildlife and Fisheries and Coastal Protection and Restoration Authority of Louisiana (LDWF Award #21-0110), funding through the support of the U.S. Geological Survey Louisiana Cooperative Fish and Wildlife Research Unit, and LUMCON star-up funds. Any use of trade, firm, or product names is for descriptive purposes only and does not imply endorsement by the U.S. Government. The funders had no role in study design, data collection and analysis, decision to publish, or preparation of the manuscript.

## Grant Disclosures

The following grant information was disclosed by the authors:

Louisiana Department of Wildlife and Fisheries and Coastal Protection and Restoration Authority of Louisiana: #21-0110.

U.S. Geological Survey Louisiana Cooperative Fish and Wildlife Research Unit.

LUMCON.

## Competing Interests

The authors declare that they have no competing interests.

## Author Contributions

- Finella M. Campanino conceived and designed the experiments, performed the experiments, analyzed the data, prepared figures and/or tables, authored or reviewed drafts of the article, and approved the final draft.
- Stephanie K. Archer conceived and designed the experiments, performed the experiments, authored or reviewed drafts of the article, and approved the final draft.
- Jillian C. Tupitza performed the experiments, authored or reviewed drafts of the article, and approved the final draft.
- Cassandra N. Glaspie performed the experiments, authored or reviewed drafts of the article, and approved the final draft.
- Megan K. La Peyre conceived and designed the experiments, performed the experiments, authored or reviewed drafts of the article, and approved the final draft.

## Animal Ethics

The following information was supplied relating to ethical approvals (*i.e.*, approving body and any reference numbers):

This study was performed under the auspices of Louisiana State University Agricultural Center protocol # A2021-08.

## Field Study Permissions

The following information was supplied relating to field study approvals (*i.e.*, approving body and any reference numbers):

Louisiana Universities Marine Consortium (LUMCON).

## Data Availability

Code and raw data are available in the Supplemental Files.

## Supplemental Information

Supplemental information for this article can be found online at http://dx.doi.org/10.7717/peerj.19346#supplemental-information.

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
