# Peer review of "Comparison of two benthic assemblage sampling gears for use on intertidal oyster reefs in Louisiana"

_PeerJ, doi:10.7717/peerj.19346_

## Round 0.1 · original submission · Major Revisions

In particular the main concerns to be addressed are related to the statistical analysis and to some data that are not presented in the results (i.e shell height).
Please address all the reviewers comments and suggestion to improve the manuscript.

·

Basic reporting

• Clear and unambiguous, professional English used throughout.
Clear and correct English language was used throughout most of the manuscript, with only few minor exceptions and redundancies that are mentioned below:
o Lines 35-36: This sentence is not clear and it could be rephrased.
o Lines 37-41: Avoid first-person pronouns and use passive voice. (Please apply this comment to the whole manuscript)
o Lines 40-41: This sentence could be combined with the previous one. Also, no need to mention the date here.
o Lines 42-48: Please add the main data of this research to the Results section of the Abstract.
o Line 61: I suggest to place ‘hereafter gear’ and the ‘literature references’ between separate brackets.
o Line 62: Consider to add ‘Moreover’ to the beginning of this sentence, to stress how the use of different gear may increase the variability of biodiversity measurements, which are already inconsistent due to the complexity of estuarine habitat.
o Line 67: Remove ‘also’.
o Line 68: Which ‘examples’? Either remove this or articulate it.
o Line 76: Edit ‘collect’ with ‘allow the collection of’.
o Lines 87-88: This sentence could be rephrased to avoid first-person pronouns – ‘In this study, a comparison between the efficiency of trays and suction samplers in monitoring four common metrics of benthic biodiversity was performed on three natural intertidal C. virginica oyster reefs’.
o Line 88: Avoid first-person pronouns.
o Lines 90-92: This sentence could be rephrased to avoid first-person pronouns – ‘To test this hypothesis, taxa abundance, richness, Pielou’s evenness and Shannon-Wiener diversity were compared between samples collected by trays and suction samplers across the three oyster reefs’.
o Line 98: Write mean and SE value in the following format – ‘mean ±SE’. (Please apply this comment to the whole manuscript)
o Lines 118 & 126: Use same character for the ‘x’.
o Lines 172-173: Please specify values. Also, change ‘substrate’ into ‘substrate volume’.
o Line 185: Add ‘across reefs’.
o Lines 197-200: This sentence could be split in half and rephrased to be clearer. Here a suggestion – ‘In this study, the efficiency of trays and suction samplers for the monitoring of four biodiversity metrics was investigated and compared on different oyster reefs. Significant differences in all biodiversity metrics were found between gear and across reefs.’
o Lines 256-260: I would consider to move this section above (line 238) for continuity, since it is related to differences between reef sites.

• Literature references, sufficient field background/context provided.
The Introduction is overall comprehensive, providing context on the research topic. However, some parts could be improved by adding additional background information. As an example, the variability of the efficiency of different sampling gear in relation to habitat characteristics and complexity is only mentioned but not thoroughly presented. Moreover, some concepts, such as the biodiversity associated specifically to oyster reefs, could use more details and references. The background section of the Abstract would also need some improvements. More details in the following comments:
o Lines 29-31: This sentence should outline the importance of biodiversity assessment for the estimate of habitat loss and monitoring of restoration/conservation progress. Please expand this concept. Also, consider splitting this sentence into two shorter ones.
o Line 32: Uncertainty of what? How does the discrepancy in the biodiversity measurements affect restoration activities and management practices? Please clarify these important implications.
o Lines 34-36: Why do different gear types produce different outcomes in terms of biodiversity measurements?
o Line 57-58: This sentence needs a citation of literature references.
o Line 59: What is this ‘highly variability’ of estuarine habitat referred to? Is this, by any chance, referred to variability over time (e.g., seasonality) or space (e.g., regional or latitudinal differences)?
o Line 61: I suggest changing the ‘efficiency of sampling gear’ into ‘consistency of measurements/results produced by the use of these gear’? The use of the gear remains efficient, but it could provide inconsistent results.
o Lines 65-66: I recommend adding more background information on the biodiversity associated to oyster reefs (with corresponding references), because of the crucial importance of these ecosystem engineers in providing habitat for other species.
o Lines 68-72: This whole section needs citations of literature references.
o Line 72: And gear that could also promote more consistent biodiversity assessments across habitats, locations, seasons, etc.?
o Line 85: Please add a few words to clarify how noise disturbance can affect biodiversity measuments.

• Professional article structure, figures, tables. Raw data shared.
The structure and format of the whole manuscript is conformed to the journal’s guidelines and standards. All raw data necessary for the review of this paper have been submitted and made available in accordance with the journal’s Data Sharing policy. Figures are relevant, high quality, well labelled & described. Only few comments to address:
o Figure 1: The resolution of B) image is too poor. Please reload a better quality version of this image. Regarding the figure caption, it is too long. Consider to shorten it (for example, there is no need to mention the trays were 1m apart as it is clear from the image).
o Figure 2: Consider to remove ’sampling method’ from the figure and leave only the legend. Regarding the figure caption, add the specific biodiversity metrics (e.g., A. taxa abundance, etc.) to the description.

Self-contained with relevant results to hypothesis.
I believe this article is self-contained with interesting results that support the stated hypothesis. However, more effort should be put to link the research questions to the findings and their related implications.

Experimental design

• Original primary research within Scope of the journal.
Confirmed.

• Research question well defined, relevant & meaningful. It is stated how the research fills an identified knowledge gap.
Knowledge gaps, research questions and hypothesis are mentioned in the Introduction, although some additional background information specifically on sampling gear and the corresponding knowledge gaps could make the rationale of this study more comprehensive. However, it is not completely clear how this research aims to fill these gaps. I recommend adding a few sentences in the Introduction (either between lines 86 and 87, or at the end of the paragraph in line 92) to clarify the overall goal and relevance of this study.

• Rigorous investigation performed to a high technical & ethical standard.
The experimental design, methodology and data analysis used in this research are satisfactory. Only few comments to address:
o Line 113: Specify what type of characterisation was performed.
o Line 114: Specify what type substrate was the reef substrate. Make sure this description matches the results in lines 170-173 (e.g., live and dead oysters).
o Lines 113-117: Also, clarify how were these reef substrate’s samples collected (e.g., scuba diving) and if the reefs where exposed or submerged during sampling.
o Line 129: Why 10 seconds? Maybe add some literature references.
o Line 139: What does this value represent? ‘0.0001g’
o Line 141: The Equation in appendix is not clear. Please review it and add some clearer description.
o Line 150: Clarify this value represented the surface area in m2 of each tray/suction sampler. ‘0.2304’
o Lines 157-159: I recommend to move this section up to line 151, and maybe add some equations of all four metrics to better describe them.
o Lines 160-162: I recommend to move this section up to line 151.

• Methods described with sufficient detail & information to replicate.
There are only some missing details/information. Please address the following comments:
o Line 103: Add GPS coordinates of the three reef sites.
o Line 109: Add GPS coordinates of the LUMCON station.
o Lines 110-111: Could you please provide information on the depth range for each reef site at each sampling time?
o Line 118 & 128: A schematic or picture of the trays and suction samplers would make these descriptions clearer.
o Lines 127-128: Please provide the model of the suction sampler device.
o Lines 134-148: This whole section needs some literature references related to the drying process and biomass calculation.
o Line 154: Clarify which normality test was used.
o Line 162: Are the values in Table S1 biomass data? Please specify this. Also, this Table does not support the information described in the text. Maybe make a new Table including both biomass and abundance data.
o Line 163: Please specify the level of statistical significance considered in this study (ɑ value).

Validity of the findings

• Impact and novelty not assessed. Meaningful replication encouraged where rationale & benefit to literature is clearly stated.
Despite the lack of novelty, this study is not redundant. The results of this research inform on variability and lack of consistency between biodiversity measurements acquired with two commonly used gear in different oyster reefs, and this information can contribute to the establishment of more effective biodiversity monitoring programs in the future. However, this important implication of this research are not well stated in the Discussion/Conclusion. I recommend addressing this important point.

• All underlying data have been provided; they are robust, statistically sound, & controlled.
Good quality data was produced and provided in this study, with only few comments to address. However, some data is missing. In particular, shell height (mm) of live oysters measured during the characterisation of reefs (line 116) was not presented in the results with the other parameters. More importantly, results of the gravimetric analysis of biomass described in details in the lab analysis section (lines 134-148) were not presented, as well as the correlation analysis between abundances and biomasses, briefly mentioned in the statistical analysis section (lines 160-162). More details in the following comments:
o Line 170-172: Data on shell height (mm) not presented. Make sure these results match the methods described in lines 114-117.
o Line 172, 260: Add unit measures to the values in Table S3. Also, it is not clear what are the values in brackets. Please clarify this.
o Line 174: No data on biomass presented. Please add some results from the gravimetric analysis of biomass described in lines 134-148. Also, add some information on the correlation analysis performed between abundance and biomass, mentioned in lines 160-162.
o Line 174, 178, 181, 183, 184, 188, 189, 191, 193: Specify these are ‘significantly’ higher.
o Line 190: Specify values from Kruskal Wallis test (Pielou’s evenness for trays between reefs).
o Line 191: Specify values from Kruskal Wallis test (Shannon-Wiener diversity at reefs 1 and 2 between gear).
o Line 193: Specify values from Kruskal Wallis test (Shannon-Wiener diversity for both gear between reefs).

• Conclusions are well stated, linked to original research question & limited to supporting results.
I acknowledge the good work done on the meta-synthesis and the structured comparison of the present results with previous studies from the literature. However, I think the Discussion/Conclusion sections lack of clear links to the original research questions. Also, some important conclusions/assumptions are not stated in this final section. Please address the following detailed comments:
o Line 196-197: This first part of the Discussion should be expanded with more details, to create a better link between the findings and the rationale of this study described in the Introduction.
o Lines 206-208: I assume these other studies were conducted using trays as well. If so, please clearly state this in the text.
o Lines 212-214: Why did you compare your results with other studies which did not consider smaller taxa? Did you find any relevant studies that analysed smaller taxa like yours in the literature? If so, consider adding these to the Discussion section.
o Lines 215-217: Why did you use a different set up compared to other studies found in the literature? Also, would you consider in future studies to investigate further the impact of deployment set up on the efficiency of tray for this type of monitoring?
o Lines 221-222: Is this information coming from the literature or the present study? Please clarify this. Also, this sentence is contradictory considering you stated the opposite in lines 198-200 (differences between gear).
o Lines 224-225: I assume these other studies were conducted using suction samplers as well. If so, please clearly state this in the text.
o Lines 228-229: Similar comment as before: Why did you use a different set up compared to other studies found in the literature? Also, would you consider in future studies to investigate further the impact of suction times and areas on the efficiency of suction samplers for this type of monitoring?
o Lines 238-241: Were these completely different/separate groups of taxa? Consider to make and add a MDS graph, as it would clearly show the differences in taxa composition between not only sampling gear but also reefs. Also, are you referring to all reefs? If so, please clarify this.
o Lines 242-243: Are these values referred to all reefs? Please clarify this.
o Line 246: Are there any studies in the literature that deployed trays for a longer time? If so, please mention them here for comparison.
o Lines 256-260: Taking into consideration that the three reefs in this study were all located in the same area, I agree with the authors that the differences in biodiversity metrics between sites might be due to local reef differences, such as structural complexity. However, this remains an assumption. Please consider to test the correlation between the parameters measured during the reef characterisation (reef substrate volume, cluster volume, abundance and shell height of live oysters) and the biodiversity metrics (taxa abundance, richness, Pielou’s evenness, Shannon-Wiener)? This would help validating your assumptions.
o Lines 266-268: Outline the potential reasons for these findings. Did you find any similar results from studies in the literature?
o Lines 275-291: The Conclusion section does not summarise in a clear and satisfactory way the main findings of this research. In my opinion, it also lacks important conclusions and implications for future studies. Please consider re-writing this section, trying to outline how this study addressed all research questions (variability between gear and sites).
Here are some tips based on the present study and the comparison with other studies, also including future considerations:
- When comparing the two sampling gear – differences were found between them in terms of all biodiversity metrics, leading to the conclusion that comparison between measures collected using different gear may be misleading. However, while more individuals and taxa, as well as dominant taxa, were found in trays, more unique taxa were collected with the suction samplers, as well as more sessile organisms. Therefore, could these methods be considered complementary? Should the combined use of the sampling gear be considered in future studies?
- When considering the same sampling gear, either trays or suction samplers – potential influence of set up (e.g., different deployment layout for trays or suction time for suction samplers), environmental factors (e.g., differences between different geographic areas), and reef characteristics (e.g., different structural complexity) on biodiversity metrics. Should these aspects of biodiversity’s monitoring techniques be explored in future studies?

Additional comments

This paper is interesting and presents useful insights on the variability and lack of consistency of biodiversity measurements acquired with different sampling gear across different environmental conditions. I believe the data provided in this study have the potential to help researchers and practitioners to establish more effective biodiversity monitoring programs for oyster reef restoration and conservation activities. It is well-written and it gets straight to the point. Thanks to the authors for their concise writing. I can tell the authors put a lot of work into the experiment itself, and I can appreciate the long hours that took to analyse all the samples. I do, however, have some concerns mainly about the Introduction section, which could use more background information to support the rationale of the experiment, as well as the Discussion section which lacks clear links to the research questions. Also, the potential future implications of this study are really important and they should be more thoroughly explored and outlined. Finally, although the authors spent time and resources to analyse the biomass of their samples, this dataset was not presented and discussed in the paper.

I recommend this paper proceed to publication after a thorough readthrough to address some major and minor comments. The suggestions herein are to help this work shine so that others can fully understand and appreciate the authors findings. Thank you for the opportunity to review.

·

Basic reporting

The language is clear, grammatically correct and professional, with some minor editorial errors. The paper is appropriately structured, with adequate background and context, and appropriate experimental design to accomplish the study objectives. The statistical analysis is correct, but a more in depth community comparison could enhance the results and discussion. The literature cited is appropriate. Some of the tables could benefit from improved formatting. I have made some specific comments in the annotated PDF attached.

Experimental design

The experimental design is appropriate for the study question, which is clearly defined. The scope of the paper is somewhat limited – it pertains to a very specific (albeit important and widespread) habitat type and sampling approaches, so the paper may not have a high impact. The methods could benefit from additional details. Refer to attached PDF for specific recommendations

Validity of the findings

This is a very simple paper with a very well defined question, and methods and analysis that clearly address the question. The discussion could be improved with a paragraph on the value and implications of the findings. The authors mention management applications of their research briefly but do not elaborate on how their study might inform monitoring programs and/or benefit management. I realize the impact and novelty of the study is not evaluated as part of the review, but I think this paper could be more interesting with some additional analyses and discussion. See attached PDF for specific comments.

Additional comments

Supplemental tables S1-3have no titles.

Reviewer 3 ·

Basic reporting

The authors present a well-written manuscript that provides adequate background and context for the study. The tables and figures are clear and appropriate. The data is shared. The results and discussion are relevant to the initial questions and there are not efforts to conflate the results. Overall, I found this a nice, concise manuscript that addresses an important challenge in sampling complex habitats.

Experimental design

The authors clearly state how the research presented fills knowledge gaps in oyster reef sampling literature. The experimental approach is sound, but the authors could provide more details or context, specifically regarding how the suction sampling occurred. Given this is a relatively uncommon practice for oyster reefs, it is unclear how exactly the samples were collected, so there is an opportunity for more details in the methods:

How was the throw trap deployed?
Was it forced down into the mud or what happened when it landed on shells so that the bottom wasn't sealed?
Were all the live oysters and shell materials collected in the sample? If yes, did the authors quantify the shell material (# of live oysters, volume, etc.)?

The last question I think is critical since the trays were standardized (i.e. at the same spots along a reef, with the same volume of shell in each tray, etc), whereas the suction samples were haphazard. That could impact the comparisons being made.

Validity of the findings

As above, I think this is a manuscript that provides important information addressing a knowledge gap in the literature about sampling complex oyster reef habitats. I have some concerns about the findings that are related to the lack of detail regarding suction sampling. Specifically, like mentioned in the experimental design, the trays were standardized in terms of both location and shell volume, where as suction samples were haphazard with no standardization for shell volume. This makes comparisons challenging. It is perhaps not surprising that suctions samples then had a higher variance in metrics examined, likely because abundance, richness, diversity, are all linked to complexity. If the authors quantified shell volume from the suction samples, this could be addressed. If not, I think that's ok but it needs to be presented and addressed as a limitation of the current study.

This is all to say that there may be some sampling-related artefacts that could affect the conclusions being drawn, and without a better explanation of the suction sampling, it is impossible to ignore those potential artefacts.

Overall, I think this is a good study addressing a knowledge gap and presents useful and unique data, but the issues mentioned above need to be addressed, either by reinterpretation with additional data (if it was collected) or incorporated into the discussion as caveats to the provided results and conclusions.

---

## Round 0.2 · accepted · Accept

The author addressed all the reviewers' comments, and the manuscript is now improved and ready for publication.

·

Basic reporting

The authors addressed all comments, also clarifying their reasons for not making some of the edits suggested.

Experimental design

The authors addressed all comments, also clarifying their reasons for not making some of the edits suggested.

Validity of the findings

The authors addressed all comments, also clarifying their reasons for not making some of the edits suggested.

Additional comments

I thank the authors for the huge effort addressing all my comments. I am aware they were numerous. I believe the manuscript has been largely improved and I enjoyed reading it again. I think this work is ready to be shared with the scientific community.

·

Basic reporting

No comment

Experimental design

No comment

Validity of the findings

No comment

Additional comments

This manuscript meets the criteria for publication and I recommend it is accepted.

Reviewer 3 ·

Basic reporting

As I mentioned in my previous review, I found this to be a well-written manuscript that addresses an important challenge in sampling complex habitats. I also appreciate that the authors made considerable efforts to address all the reviewer concerns and make updates to the text as necessary.

Experimental design

Although I still have some, we'll call them philosophical, concerns about the suction sampling design and what it means for the comparisons, the authors did a nice job at providing more detailed explanation of the technique to address concerns by myself and other reviewers. I think that the design is sufficient and relevant for their questions and am ok with the explanations they provided in the rebuttal.

Validity of the findings

See above, I think the authors did a nice job here addressing the reviewer concerns and making changes to the text and discussion.